# EDU-RAG: A RAG Benchmark with Web-enhanced Content in Education Domain. Can RAG Help AI Tutor?

## ABSTRACT

Hallucination has been a persistent challenge when using Large Language Models (LLMs). Retrieval-Augmented Generation (RAG) has emerged as a popular approach to mitigate this issue by maintaining context and coherence in generated outputs, as well as incorporating customized knowledge (Lewis et al., 2020a). In this paper, we propose a benchmark dataset for evaluating LLM performance in the domain of middle-school science question answering, using textbook questions augmented with real-world web search results. We assess the performance of various LLMs, including GPT-4o, Llama2-7b, and Llama3-8b, with and without the application of RAG. Our goal is to determine whether RAG can reduce hallucinations stemming from the inherent biases of pre-trained LLMs or from the retrieval of irrelevant knowledge, even when relevant information is accessible (Ji et al., 2023a). The dataset and methodology introduced here provide a robust foundation for advancing the evaluation and development of RAG techniques in mitigating hallucinations across diverse LLMs.

## 1 INTRODUCTION

Recent research in RAG emphasizes its potential to augment natural language generation by retrieving relevant documents or information, which helps maintaining context and coherence in generated outputs (Lewis et al., 2020a). A notable challenge highlighted by research is hallucination – RAG models may draw invalid correlations or rely on biased model knowledge, overriding retrieved information and subsequently hallucinating, even when relevant information is accessible. (Ji et al., 2023a).

Our project aims to address these challenges by refining RAG benchmarks and algorithms. Addressing hallucinations remains a key concern in RAG, requiring advanced techniques to evaluate information relevance and factual consistency effectively. Furthermore, domain-specific nuances may be incorporated to reduce misleading or inaccurate outputs, particularly in critical areas such as education.

Education domain is selected because Artificial Intelligence (AI) tutors are popular applications for LLMs. We would like to study the current status of the field, and provide a useful benchmark to help improve AI tutors. This is an important field because AI tutors can provide low cost solutions to people who don't have access to expensive education resources.

Figure 1 demonstrates an example of the benchmark. The input contains a questions, choices and the expected answer. This is a question related to Mendel's experiments in the biology field. With the LLM (Llama2-7b in this case), it provided a wrong answer. With the RAG solution (Llama2-7b with the input of web search results), it provided the correct answer. The benchmark also contains web search results based on the questions, which can be used for the RAG algorithm.

The project is inspired by Meta KDD Cup 2024 RAG: Comprehensive RAG Benchmark (CRAG) (Yang et al., 2024). Currently CRAG only includes five domains: Finance, Music, Movie, Sports, and Open, and it does not include the Education domain. Our project creates a new benchmark for the Education domain, with different data sources and processes.

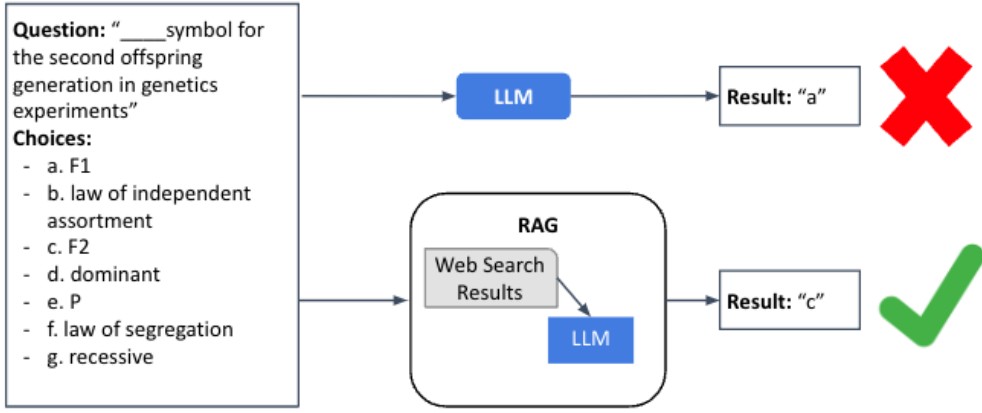

Figure 1: Example of Question, Choice Set, and Result with LLM with and without RAG

## 2 RELATED WORK

RAG has significant potential to augment natural language generation by retrieving relevant documents or information, which helps in maintaining context and coherence in generated outputs (Lewis et al., 2020a). There is an also a more recent survey shows that RAG models may draw invalid correlations with retrieved knowledge or use biased model knowledge, despite access to relevant information (Ji et al., 2023a).

The Head-to-Tail Paper (Sun et al., 2024) offers valuable insights into LLMs' knowledge capabilities, providing a foundation for further research on enhancing factuality and reducing hallucinations in NLP models. In addition, there are methods like FreshLLMS to enhance factual consistency in LLMs through real-time information retrieval, offering a holistic view of NLP research in reducing hallucinations and improving factuality (Vu et al., 2023).

LLMs are very good at memorizing the content used for its pretraining. However, there are many drawbacks of using its memorization capability directly for Question Answering (QA) tasks. For example, depending on the size of the model, the quality of the pretraining data, and the type of questions, LLMs' memorization capability can be very limited and hard to be controlled. The LLMs are also hard to be updated unless through retraining or fine tuning, so LLMs can not handle questions on recent events if deployed. Arguably the most problematic drawback of using LLMs for QA is it can hallucinate, especially so if the models do not know if the context contains the information needed to answer a question.

Fine tuning is a straightforward way to update the knowledge of LLMs if the updated information become available. However, due to the scarcity of GPUs and limitations of access to high quality data, it is not feasible for lots of use cases. Furthermore, the behavior of tuning language model is not well studied, and fine tuning with bad data or practices might decrease the model's capabilities, causing modality collapse unless more complex methods such as RLHF are used (Ouyang et al., 2022).

Retrieval-Augmented Generation (RAG) (Lewis et al., 2020b; Gao et al., 2024) is a popular way to address the shortcomings of LLMs, by augmenting LLMs with non-parametric data sources, utilizing LLMs' powerful in-context learning (Brown et al., 2020) capability. (Gao et al., 2024) grouped the approaches of using RAG for LLMs into three categories: Naive RAG, Advanced RAG (Ma et al., 2023; Zheng et al., 2024), and Modular RAG (Yu et al., 2023; Shao et al., 2023). The Naive and Advanced RAG are widely in practice due their simplicity and low cost on development. They are generally consisted of three parts: non-parametric database curation, retrieving relevant snippets from the database given the query, LLMs generation by in-context learning and prompt engineering with the snippets that are related the query.

While the research on combining LLMs and RAG for QA mainly focus texts, there is also research on utilizing resources beyond pure texts, such images (Chen et al., 2022), audio (Zhao et al., 2023), video (Yang et al., 2023) and code (Nashid et al., 2023) to augment the capabilities of language models.

There are various efforts on creating RAG benchmarks and proposing appropriate evaluation metrics in recent years. (Yang et al., 2024) is one of the recent ones, which is the foundation of this paper. (Yang et al., 2024) created a factual question answering benchmark of 4,409 question-answer pairs and mock APIs to simulate web and Knowledge Graph (KG) search. It also proposes an evaluation mechanism, which distinguishes hallucination and missing answers, and gives hallucination a higher penalty. (Chen et al., 2024) created a RAG evaluation benchmark in both English and Chinese, and analyzed different LLMs from 4 aspects: noise robustness, negative rejection, information integration, and counterfactual. They found that LLMs show a certain degree of noise robustness, but struggle significantly in other aspects.

Apart from specific RAG datasets, there are many existing Question-answering (QA) datasets which include context passages for each question. These datasets can also be used for RAG experiments, and cover a wide range of questions such as multiple-choice QA (Pang et al., 2022; Clark et al., 2018; Talmor et al., 2019), single-hop QA (Kwiatkowski et al., 2019; Joshi et al., 2017; Rajpurkar et al., 2016), multi-hop QA (Yang et al., 2018; Ho et al., 2020; Trivedi et al., 2022), and Domain-specific QA (Dasigi et al., 2021; Möller et al., 2020; Wang et al., 2024).

Hallucination (Ji et al., 2023b) in LLMs is a well-known problem, and can generally be grouped into intrinsic hallucinations and extrinsic hallucinations. Intrinsic hallucinations are generated output that contradicts the input content. For example, generated abstractive summarization that includes contradicting facts from the original article. Another example is generated answers contradicting facts presented in the context of a RAG task. Extrinsic hallucinations include generated output that can't be verified from the input. In this situation, the LLMs draw upon their stored "knowledge" and produce outputs that can be correct and relevant, correct and not relevant, or erroneous. Hallucination in LLMs is considered a very challenging issue which can not be improved by directly scaling up models (Lin et al., 2022).

One way to deal with hallucinations is detection. (Snyder et al., 2023) trained a binary classifiers on artifacts from the LLM's inputs, internal state and outputs to classify the generated outputs into hallucinations and non-hallucinations. (Kadavath et al., 2022) fine-tuned the model itself to predict whether it knows the correct answer. (Azaria and Mitchell, 2023) trained a classifier on the LLM's internal state to predict whether the output is truthful. (Zhang et al., 2024) invested using the LLMs to self-identify hallucination after the answers are generated.

## 3 APPROACH

Our approach comprises two primary components. First, we constructed a benchmark dataset tailored for LLM RAG use cases in the education domain. This dataset includes middle school-level science question-answering (QA) pairs in multiple-choice format, along with an augmented knowledge base for each question derived from web search results. Second, we implemented a RAG module and evaluated the performance of the LLMs with RAG by incorporating the augmented knowledge as in-context learning in the prompt, to be compared with the performance of vanilla LLM models.

### 3.1 BENCHMARK CONSTRUCTION

The benchmark construction process extracted QA pairs from an existing Textbook Question Answering (TQA) dataset(Kembhavi et al., 2017), and added the additional web content data per question. As shown in Figure 2, the Web Content Handler calls the Web Search Engine API to get the top Uniform Resource Locators (URLs) based on the question, and then crawls the web content from the URLs. More details are as follows.

**QA Pair**: We leveraged the TQA dataset (downloadable from https://allenai.org/data/tqa), which draws textbook content such as lesson topics, and questions and answers from middle school science curricula such as Life Science, Earth Science, and Physical Science textbooks. We sampled from the 13,693 non-DiagramQuestions as example in Appendix 4 that don't reply on accompanying diagrams

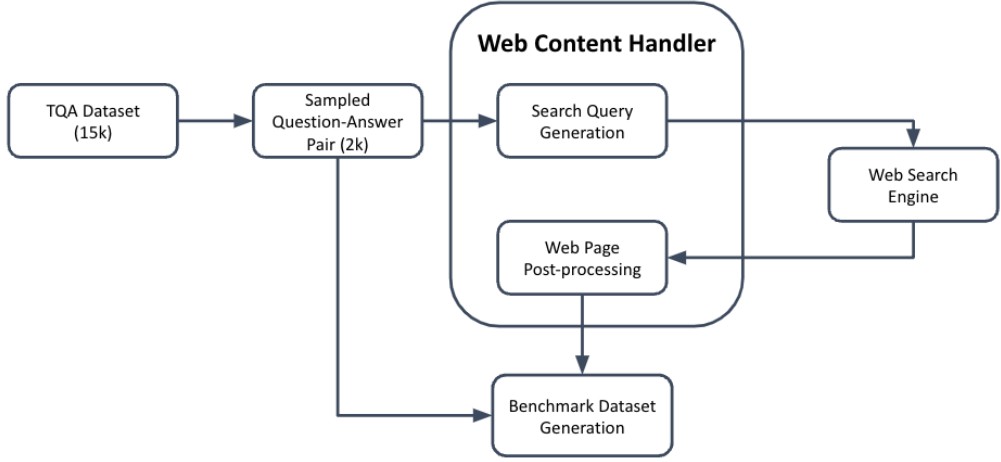

Figure 2: Benchmark Construction Process

including multiple-choice, including true/false questions also in multiple-choice format. We have the dataset in csv with columns of beingAsked (question), answerChoices (choice-set) and correctAnswer (correct choice label).

**Web Page Content**: For each question, we leveraged the Google Search Python API googlesearch-pythonweb to pragmatically obtain the URLs of the top search results by querying the question prompt (not including the choice set), The cleaned article content from these web pages was then extracted using web scraping services (beautifulSoup, requests) and natural language toolkit (nltk, newspaper3k). We converted each web page content into a txt file. The setup was configured to fetch the top 10 Web pages for each search query, and with a request timeout of 5 seconds to avoid issues related to excessive website redirects or delay due to login/human verification. Web page content files smaller than 0.5K were filtered out, as they typically contained web access errors or empty frames. This configuration allowed us to obtain an average of three web pages per question, with the average length of the extracted content being approximately 8,285 characters, as detailed in Appendix Table 4 4.

## 3.2 RAG DESIGN

As shown in Figure 3, the key components of the RAG design include Retriever, Reranker, Generator and Evaluator.

**Retriever**: Pre-processes the extracted content for each question, including concatenating the content from all web pages fetched for one question, and transforming the article into sentences.

**Reranker**: Selects the top K closest sentences for each question, by encoding each question and each sentence from the extracted web search content to vector representations via S-BERT Sentence Transformer (Reimers and Gurevych, 2019). After that, computes the similarity scores between each sentence and the question, and picks the top K.

**Generator**: Calls the LLM Service, instructing it to respond with only the one choice letter, with the augmented sentences in the prompt (reference to the Llama Prompt Structure without RAG in Prompt 3.1, Llama Prompt Structure with RAG in Prompt 3.2 and GPT-4o Prompt Structure with RAG in Prompt 3.3).

**Evaluator**: Calculates the evaluation metrics.

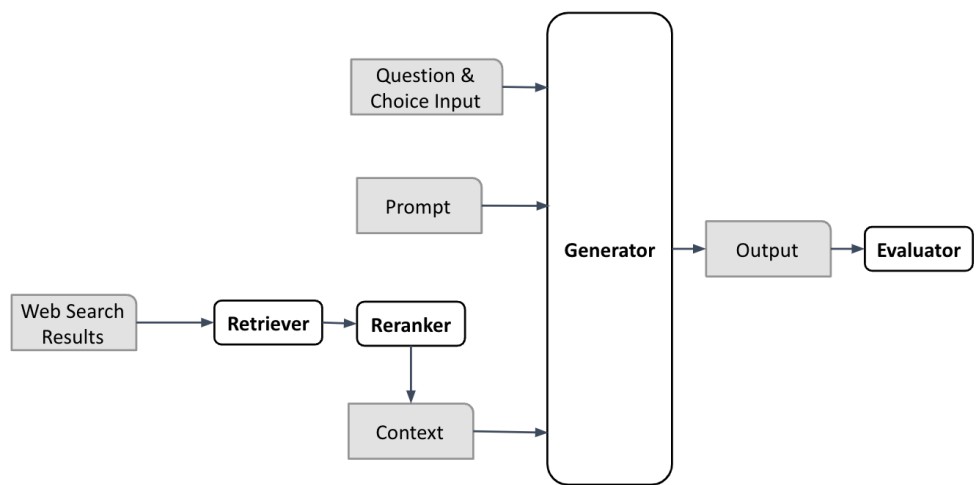

Figure 3: RAG Design

---

**Prompt 3.1: Llama Prompt Example without RAG**

- **Instruction Part1:** You are a helpful science tutor. You should output only a single letter indicating the choice you make. You must choose a letter even if you are unsure.
- **Question Part:** <TQA Dataset Question> e.g., _____symbol for the second offspring generation in genetics experiments
- **Choices Set part:** <TQA Dataset Choice Set for the Question> e.g., a. F1 | b. law of independent assortment | c. F2 | d. dominant | e. P | f. law of segregation | g. recessive
- **Instruction Part2:** I have to choose a letter option. The letter chosen is

---

**Prompt 3.2: Llama Prompt Example with RAG**

- **Instruction Part1:** You are a helpful science tutor. You should output only a single letter indicating the choice you make. You must choose a letter even if you are unsure.
- **Knowledge Augmentation:** <Closest sentences from top search results:> e.g., The parent plants in the experiments are referred to as the P (for parent) generation. You can explore an interactive animation of Mendel's first set of experiments at this link: http://www2.edc.org/weblabs/Mendel/mendel.html. Reciprocal crosses generated identical F 1 and F 2 offspring ratios. By examining sample sizes, Mendel showed that traits were inherited as independent events. In modern terms, we call the parents the P generation and the offspring the F1 generation. F1 stands for first filial generation – filial means child. For the example of the P generation of purple crossed with white flowers, all the F1 offspring were purple. If he crossed plants that varied in a single character – purple flowers and white flowers, for example – he found that the first generation of offspring always matched the appearance of one of the parents. But the F 1 possesses the information needed to produce both parental phenotypes in the following generation. If two F1 offspring were crossed, though, both traits of the original parents would be seen in the next generation, called the F2 generation. For the purple flower x white flower cross, there were both purple and

> white F2 offspring. Mendel called purple the "dominating" trait; in modern terms we use the word dominant....
>
> - **Question Part:** <TQA Dataset Question> e.g., ____symbol for the second offspring generation in genetics experiments
>
> - **Choices Set part:** <TQA Dataset Choice Set for the Question> e.g., a. F1 | b. law of independent assortment | c. F2 | d. dominant | e. P | f. law of segregation | g. recessive
>
> - **Instruction Part2:** I have to choose a letter option. The letter chosen is

---

**Prompt 3.3: GPT-4o Prompt Example with RAG**

- **Instruction Part:** Assume you are a scientist. Given the multi-choice question below, generate the correct answer. You must output only the letter of the correct choice within (a, b, c, d, e, f, g). If you don't know, answer "I don't know". Please be concise and answer in one word.

- **Knowledge Augmentation:** <Closest sentences from top search results:> e.g., The parent plants in the experiments are referred to as the P (for parent) generation. You can explore an interactive animation of Mendel's first set of experiments at this link: http://www2.edc.org/weblabs/Mendel/mendel.html. Reciprocal crosses generated identical F 1 and F 2 offspring ratios. By examining sample sizes, Mendel showed that traits were inherited as independent events. In modern terms, we call the parents the P generation and the offspring the F1 generation. F1 stands for first filial generation – filial means child. For the example of the P generation of purple crossed with white flowers, all the F1 offspring were purple. If he crossed plants that varied in a single character – purple flowers and white flowers, for example – he found that the first generation of offspring always matched the appearance of one of the parents. But the F 1 possesses the information needed to produce both parental phenotypes in the following generation. If two F1 offspring were crossed, though, both traits of the original parents would be seen in the next generation, called the F2 generation. For the purple flower x white flower cross, there were both purple and white F2 offspring. Mendel called purple the "dominating" trait; in modern terms we use the word dominant....

- **Question Part:** <TQA Dataset Question> e.g., ____symbol for the second offspring generation in genetics experiments

- **Choices Set part:** <TQA Dataset Choice Set for the Question> e.g., a. F1 | b. law of independent assortment | c. F2 | d. dominant | e. P | f. law of segregation | g. recessive

---

In the Retrieval stage, we leverage cosine similarity. For each question i in the 1.25K QA pair, we generate the question sentence embedding $\mathbf{Q_i}$. For each sentence j in the retrieved documents via the top 10 web search result pages for question i, we generate the embedding $\mathbf{D_{ij}}$. The cosine similarity is calculated as follows And then for each question i, fetch the top-k sentences based on these similarity scores.

$$\mathrm{sim}_{\mathrm{cos}}(\mathbf{Q_i}, \mathbf{D_{ij}}) = \frac{\mathbf{Q_i} \cdot \mathbf{D_{ij}}}{\|\mathbf{Q_i}\|\|\mathbf{D_{ij}}\|} \tag{1}$$

## 4 EXPERIMENTS

### 4.1 DATA

We tested different models on a sample of 1,255 TQA question-answer pairs, including the question, choice set, and augmented knowledge in-context for RAG use case.

## 4.2 EVALUATION METHOD

To simplify measurement, as TQA provides a multiple-choice set, we asked the LLM to return a single choice letter rather than a natural language response. In addition, when there is case with missing response / failed prediction, or respond with "I don't know", it's considered a missing.

Since our research topic is to evaluate whether RAG can help reduce hallucination, the total score metrics is designed to penalize on incorrect predictions and to encourage the model to predict "I don't know" instead. If the model predicts "I don't know" or nothing, we treat this as "missing" cases and there is no deduction on total score. If there is a wrong prediction, we treat this as "incorrect" cases, and there is deduction on total score.

- Accuracy (**Acc.**): the percentage of correct predictions
- Miss Rate (**Miss.**): the percentage of missing predictions (e.g., "I don't know", "I can't answer...")
- Hallucination Rate (**Hall.**): the percentage of incorrect predictions
- Total Score (**Score$_h$**): sum of the scores across all predictions divided by total number of predictions. Score is defined as that for each predicted answer, 1 point if correct, 0 point if missing, and -1 point if incorrect. This will result in an additional penalty for a hallucinatory answer.

Following the conventions used in CRAG (Yang et al., 2024), we defined the Total Score as

$$\textbf{Score}_h = 1.0 \cdot \textbf{Acc.} + (-1.0) \cdot \textbf{Hall.} + 0.0 \cdot \textbf{Miss.} = \textbf{Acc.} - \textbf{Hall.}$$

## 4.3 EXPERIMENTS DETAILS

We first evaluated the performance of the vanilla LLMs without RAG on the 1,255 sampled QA pairs. Then we integrated the top sentences, which ranked and pruned web search results sentences into the prompts as in-context learning augmented knowledge, with the same model configuration.

- Common parameters used in the Retrieval Stage for all models:
  - Number of TopK closest sentences from full search results content per question = 10
  - max web search results length: 7500, to avoid exceeding token limitation for LLM models, especially when using non-pruned full documents
- Models:
  - Vanilla Llama3 8B instruction fine-tuned: meta-llama/Meta-Llama-3-8B-Instruct
  - Vanilla Llama2 7B instruction fine tuned: meta-llama/Llama-2-7b-chat-hf
  - Chatgpt-4-o
- Models Params:
  - max_new_tokens = 1
  - Minimize Randomization, e.g. do_sample=False

## 4.4 RESULTS

Experiment Results show that RAG with Pruned Web Search Contents improved the accuracy of the three LLM models. The improvements are especially significant for Llama-2, for which the RAG solution increased the accuracy rate from 25.34% to 52.03% in Table 1. The RAG solution also shows accuracy improvement in Llama-3 and GPT-4o. The Vanilla versions in the table refers to the basic version of the model without RAG.

The total score is also higher for models with RAG solution. We computed the total score based on the number of correct, incorrect, missing questions in Table 1, and then generated the Table 2 to display the total score. The total score is defined as accuracy rate - hallucination rate, which is explained in Section 4.2. The total score values the response with accurate answers, and penalize the response with wrong answers. In this way, this metrics favors the solutions with answers "I don't know" or missing results, and penalizes the solutions with wrong answers.

|  | Accuracy Rate | Number of Questions | | |
|---|---|---|---|---|
|  |  | Correct | Incorrect | Missing |
| Llama2-7b - Vanilla | 25.34% | 318 | 234 | 703 |
| Llama2-7b - RAG | 52.03% | 653 | 510 | 92 |
| Llama3-8b - Vanilla | 70.12% | 880 | 375 | 0 |
| Llama3-8b - RAG | 74.26% | 932 | 323 | 0 |
| GPT-4o - Vanilla | 87.57% | 1099 | 141 | 15 |
| GPT-4o - RAG | 88.68% | 1112 | 130 | 13 |

Table 1: Accuracy Rate (1255 Samples)

|  | Score Breakdown | | | |
|---|---|---|---|---|
|  | Accuracy Rate | Hallucination Rate | Missing Rate | Total Score (Accuracy Rate - Hallucination Rate) |
| Llama2-7b - Vanilla | 25.34% | 18.65% | 56.02% | 6.69% |
| Llama2-7b - RAG | 52.03% | 40.64% | 7.33% | 11.39% |
| Llama3-8b - Vanilla | 70.12% | 29.88% | 0% | 40.24% |
| Llama3-8b - RAG | 74.26% | 25.74% | 0% | 48.53% |
| GPT-4o - Vanilla | 87.57% | 11.24% | 1.20% | 76.33% |
| GPT-4o - RAG | 88.61% | 10.36% | 1.04% | 78.25% |

Table 2: Score Breakdown and Total Score for Benchmark Vanilla LLM Models (1255 Samples) [1]

## 5 ANALYSIS

Through the EDU-RAG benchmark and the experiments, we can conduct analysis on the following research questions (RQs):

**RQ 1**: Is this a valid benchmark to help evaluate the effectiveness of different RAG algorithms?

Yes. This is a valid benchmark to help evaluate RAG algorithms. The initial results successfully demonstrated that the dataset is valid, and can be used for the community to test different models. We also showed that a basic RAG algorithm with LLMs can improve the accuracy and total score compared with basic LLMs.

**RQ 2**: Can RAG help improve the accuracy?

Yes. The accuracy of the RAG solutions with Llama2-7b, Llama3-8b and GPT-4o is better than their basic LLM versions without RAG. For Llama2-7b, it shows 26.69% absolute level improvement. For Llama3-8b, it shows 4.14% absolute level improvement. For GPT-4o, it shows 1.1% absolute level improvement.

**RQ 3**: Can RAG help reduce the hallucination?

It depends. It reduced the hallucination rate of Llama3-8b from 29.88% to 25.74%, and GPT-4o from 11.24% to 10.36%. For some baseline models with low hallucination rate, e.g. Llama2-7b with 18.65% hallucination rate, it reduced the hallucination rate to 40.64%. Part of the reason can be that the RAG solution provided some irrelevant context, and it generated more incorrect answers. This can also be optimized by improving the RAG solutions, e.g. apply more rigorous threshold on the pruning algorithm. There are also opportunities for iterating on the RAG algorithms, if time allowed.

---

[1]Results are stored in more than two decimal points, which could cause small differences if using the three rates to calculate the scores. For example, when stored in 5-decimal, Llama3-8b RAG Accuracy Rate is 74.26295% (74.26% when rounded up) while Hallucination Rate is 25.73705% (25.74% when rounded up), which will derive Total Score to be 48.52590% (48.53% when rounded up).

## 6 CONCLUSION

We introduced EDU-RAG, a RAG benchmark with web-enhanced content in the education domain. This benchmark serves as a valuable resource for the community to evaluate different RAG algorithms and is particularly beneficial for applications in the education domain, such as AI tutors.

We also evaluated the benchmark using basic RAG solutions with recent LLMs, including Llama2-7b, Llama3-8b and GPT-4o. The preliminary results show that RAG can improve accuracy and help AI tutors. It can also help reduce hallucination, and that requires more thoughtful designs, such as handling the flow in pruning documents.

There are also many other exciting topics to explore in future work:

### 6.1 FUTURE WORK

- **Fine-tuning Stage**: Supervised Fine-Tuning (SFT) will be used to improve the model. In addition, we will construct random negative samples to address the hallucination problem.
    - SFT: Two types of fine-tuning models can be trained. The first type is to create prompts with only the question as input and construct the answer as output. The second type is to construct prompts with the question and the crawled web pages together as the input, and keep the answer as the output. The second type can be used to improve the model's capability to understand web pages.
    - Negative Samples: Negative samples are essential for reducing the hallucination. It is better to answer "I don't know", rather than a wrong answer. To enable this, we can randomly select irrelevant web pages and combine them with questions to formulate the input, and use "I don't know" as the output, which is similar to the strategy used by Adobe (Sharma et al., 2024).
- **Web Content Handler**: This stage can also be improved by leveraging other frameworks such as Scrapy. Scrapy is an extensible framework widely used in the research community. With preliminary tests, this can crawl more web pages. Due to time constraints, we did not test the LLMs with a larger dataset; however, we have already obtained a larger benchmark dataset.

## 7 ETHICS STATEMENT

- Vulnerability and Safeguarding: Minors are more vulnerable to stimuli, misinformation, and potentially harmful content. In our case, we leveraged flagship LLM models and fetched web content via top Google Search results. Both are commercial tools built on curated content and through organic content filters. Additionally, we have limited the ability to respond to multiple-choice questions (max_token = 1) to mitigate output risks. For intentional misuse (e.g., inputting unsafe content via multi-choice format and using misleading questions to provoke risky responses), more formal methods can be employed. These include implementing content filtering for both input and output to reject abusive questions or fine-tuning with human feedback, particularly from diverse, inclusive, and Responsible AI datasets such as the Meta AI Safety Benchmark (Vidgen et al., 2024). For teen use cases, we can also leverage adversarial training and red-teaming to evaluate and enhance model robustness against possible misuse or abusive scenarios.
- Academic Integrity: AI Tutors could be abused as cheating tools that provide unfair advantages to dishonest students and devalue the efforts of students who work hard to achieve success. This issue is especially concerning for minors. Our project aims more on the service-provider side to evaluate and achieve a high-performance Question Answering system that can derive the correct answer first, while in a future production phase in real customer-facing environment, the service-provider should enable Parent / Educator Control mechanisms, or product instruction such as learning intent and objective setting/monitoring.
- Privacy and Data Security: In the real-world use case of AI Tutor tools, any data collected during sessions should be handled securely and anatomized to protect privacy. In the education domain, users are often minors, which imposes stricter requirements for the collection of personal information.

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

## 8 APPENDIX

Due to time limitations, we ran RAG based on 1,255 samples, while we have also run the benchmark using Vanilla LLM models based on 2,512 samples, with results as below.

| | Score Breakdown | | | |
|---|---|---|---|---|
| | Accuracy Rate | Hallucination Rate | Missing Rate | Total Score |
| Llama2-7b | 24.56% | 18.75% | 56.69% | 5.81% |
| Llama3-8b | 70.06% | 29.94% | 0% | 40.12% |
| GPT-4o | 89.41% | 9.55% | 1.04% | 79.86% |

Table 3: Total Score and Score Breakdown for Benchmark Vanilla LLM Models (2512 QA Samples)

| | |
|---|---|
| **Average number of web pages retrieved for each question** | 3.89 |
| **Average number characters of each extracted web page content** | 8284.91 |

Table 4: Stats of Crawled Web Content

```
"NDQ_016973": {
    "answerChoices": {
        "a": {
            "idStructural": "a.",
            "processedText": "true",
            "rawText": "a. true"
        },
        "b": {
            "idStructural": "b.",
            "processedText": "false",
            "rawText": "b. false"
        }
    },
    "beingAsked": {
        "processedText": "an electromagnet has north and south magnetic poles and a magnetic field.",
        "rawText": "3. True or false: An electromagnet has north and south magnetic poles and a magnetic field."
    },
    "correctAnswer": {
        "processedText": "a",
        "rawText": "true"
    },
    "globalID": "NDQ_016973",
    "idStructural": "3.",
    "questionSubType": "True or False",
    "questionType": "Multiple Choice"
},
```

Figure 4: TQA nonDiagramQuestion Raw JSON Example

```
{
    "url": "https://beachapedia.org/Shoreline_Structures",
    "content": [
        "Why We Should Care\n\nSeawalls, groins, jetties and other shoreline stabilization structures have had tremendous impacts on our nation's beaches.",
        "Shoreline structures are built to alter the effects of ocean waves, currents and sand movement.",
        "They are usually built to \"protect\" buildings that were built on a beach that is losing sand.",
        "Sometimes they are built to redirect rivers and streams.",
        "Other times they are constructed to shelter boats in calm water.",
        "In many cases, seawalls, jetties, breakwaters and groins have caused down-coast erosion problems with associated costs that have greatly exceeded the cons",
        "Every surfrider knows that there are groins and jetties that have incidentally improved wave riding.",
        "However, in many other areas shoreline construction has ruined wildlife habitat, destroyed surfing waves and caused beaches to erode.",
        "As beach lovers and environmentalists, we need to understand the consequences of shoreline structures so that we may be able to effectively influence deci",
        "As an environmental group committed to maintaining the natural shoreline and beach equilibrium, we are usually opposed to construction that will disrupt t",
        "The Basics\n\nErosion: Where Has All The Sand Gone?",
        "Every winter, the newspapers show pictures of oceanfront buildings falling into giant surf.",
        "Beaches are not static piles of sand.",
        "Ocean currents cause beaches to move constantly.",
        "Beach sand is primarily a product of the weathering of the land (such as natural erosion of coastal bluffs).",
        "Sand can also come from ocean organisms such as coral.",
        "However, most of the sand along the world's beaches comes from rivers and streams.",
        "When natural processes are interfered with, the natural supply of sand is interrupted and the beach changes shape or can disappear completely.",
        "Sand production stops when coral reefs die from pollution, when coastal bluffs are \"armored\" by sea walls and when rivers are dammed or channelized (lin",
        "The sand that collects behind upstream dams and reservoirs is often \"mined\" and sold for concrete production.",
        "It then never makes it to the beach.",
        "A public resource essential for our beaches is instead sold for private profit.",
        "In the face of eroding beaches, owners of beachfront property will often try to use their political influence to demand that \"something be done.\"",
        "The intelligent action would be to move the building away from the ocean.",
        "Unfortunately, what has often been done in the past has been to armor the coastline with rocks, concrete and steel.",
        "This does not protect or maintain the beach - it only protects the buildings, temporarily.",
        "Millions of taxpayer dollars have been wasted subsidizing beachfront building.",
        "Federal flood insurance and expensive Army Corps of Engineer projects have done very little to make oceanfront buildings safe and have hastened beach eros",
        "In many cases, it would be more cost-effective for taxpayers to have the government buy the coastal property, condemn the buildings and allow the area to",
        "In urbanized areas with expensive real estate, a more cost effective and environmentally sound alternative to shoreline structures may be to periodically",
        "The Littoral Cell\n\nOn the West Coast of the U.S., beach sand moves from river mouths to the beach.",
        "It then moves along the coast in the direction of prevailing currents and eventually it moves offshore.",
        "This sand transport system is called a littoral cell.",
        "When waves break at an angle to the shoreline, part of the wave's energy is directed along the shore.",
```

Figure 5: Extracted Web Content Example

