# OpenReview forum: "EDU-RAG: A RAG Benchmark with Web-enhanced Content in Education Domain. Can RAG Help AI Tutor?"
_ICLR.cc/2025/Conference — ICLR 2025 Conference Withdrawn Submission_

### Official Review · Reviewer_vpbE · 2024-10-29

**Soundness:** 2
**Presentation:** 1
**Contribution:** 1
**Rating:** 3
**Confidence:** 4

**Summary:**

This paper introduces EDU-RAG, a benchmark dataset designed to evaluate the performance of Retrieval-Augmented Generation (RAG) techniques in the context of middle-school science question answering. The dataset combines textbook questions with relevant web search results, addressing the challenge of hallucination in Large Language Models (LLMs) like GPT-4o and Llama2-7b.

**Strengths:**

Originality: The paper presents a new benchmark dataset, EDU-RAG, specifically tailored for evaluating RAG techniques in the domain of middle-school science question answering.
Quality: The methodology for constructing the benchmark dataset is explained, including the selection of questions, retrieval of web content, and processing of the data.
Clarity: The authors also provide a comprehensive overview of related work, placing their contributions within the broader context of RAG research.
Significance: The EDU-RAG benchmark has the potential to significantly advance the evaluation and development of RAG techniques for educational applications.

**Weaknesses:**

- The paper only evaluates a basic RAG algorithm design, consisting of a retriever, reranker, and generator. While this serves as a useful baseline, it does not explore the effectiveness of more advanced RAG techniques, such as modular RAG or advanced reranking methods.
- The paper acknowledges that RAG can reduce hallucination in some cases but also highlights instances where it may worsen the issue. However, the analysis of this phenomenon is limited, and the paper does not delve into the underlying reasons of it.
- Typo: The title of section References appears twice.
- Missing figure: The figure 2 is missing.

**Questions:**

- Have you considered expanding the scope of evaluation to include more complex question answering scenarios, such as open-ended questions or multi-hop reasoning tasks for educational applications?
- How did you address the potential bias in the web content retrieved using Google Search?
- How did you assess the quality and relevance of the web content retrieved for each question?

---

### Official Review · Reviewer_C9X8 · 2024-11-04

**Soundness:** 1
**Presentation:** 1
**Contribution:** 1
**Rating:** 3
**Confidence:** 4

**Summary:**

This paper proposes an RAG benchmark in the education domain. The benchmark builds on top of AI2's TQA dataset, augmented with web search results for each question. The paper evaluates a few LLMs on performance on this benchmark where the metric is answer correctness (note that the author defined "hallucination" in terms of answer correctness). The author finds that RAG improves performance of LLMs in answering these questions. The author claims this benchmark will be useful for "advancing the evaluation and development of RAG techniques in mitigating hallucinations across diverse LLMs."

**Strengths:**

I like the direction that this paper attempts to go: evaluating and benchmarking RAG systems and LLMs and checking whether they hallucinate is very important. Doing all of these in the context of education is also important. I'm excited to see more work along this direction.

**Weaknesses:**

The paper seems to be poorly presented/written. Contributions seem insufficient and unclear.

In terms of presentation:
- Missing figure (figure 2)
- Table 1 and 2 are repeated
- Not sure what the example question means in Fig 1 (the question is used multiple times in subsequent results and illustrations)

In terms of contribution:
- It is unclear to me what the *fundamental* differences or advancements that the proposed benchmark has compared to prior work. For example, I'm not sure how this benchmark is different from Yang 2024's. Expanding their dataset to a new domain doesn't seem super novel to me unless there are some fundamental differences or challenges which requires innovations in either constructing the benchmark or evaluating on the data, neither of which I find novel in this paper. This is my main concern.
- The research questions seem to be studying well-known conclusions. For example, RQ2 is, roughly speaking "can RAG help improve LLM performance"? I think the answer is YES by now and it is widely known. When reading the abstract, I thought the authors would evaluate how LLMs would perform in the presence of irrelevant information (line 22 ~ 23), which I think would be a deeper and more interesting analysis than RQ2, but unfortunately the authors did not present such analyses or findings. In general, I find the conclusions and analyses in this paper either already known or shallow.
- Some of the suggested future work is already done a few years ago. For example, SFT on retrieved content is explored in this paper back in 2022: https://arxiv.org/pdf/2201.08239

Given that the paper appears not ready for presentation and the contributions are unclear and not particularly novel (see above for details), I would not recommend acceptance of this paper.

**Questions:**

What are the fundamental differences between this benchmark and previous ones? Or more specifically, what are the fundamental differences in the education domain versus finance etc in Yang 2024?

The prompt says "You must choose a letter even if you are unsure." How would one expect the LLM to output "I don’t know” if they are not instructed to do so?

What motivates the authors to define hallucination as answer correctness? The way the authors defined hallucination seems more like model's problem solving or reasoning capabilities rather than hallucination.

---

### Official Review · Reviewer_uMw2 · 2024-11-06

**Soundness:** 1
**Presentation:** 1
**Contribution:** 1
**Rating:** 1
**Confidence:** 5

**Summary:**

This paper proposes a benchmark dataset for evaluating LLMs in the education domain. Further authors experiment with RAG based approach to reduce hallucinations in LLMs in the context of education.

**Strengths:**

1. The motivation behind the research in the education domain is relevant and apt.
2. Authors create a new dataset of MCQA for middle school level science. The dataset is augmented with information obtained via web-searches.

**Weaknesses:**

1. The paper lacks novelty as authors are implementing the standard RAG architecture for the QA task.
2. The newly created dataset is mainly an extension of an existing TQA dataset augmented with content from the web, so there is very limited innovation.
3. The paper doesn't report any new findings; the authors show that RAG helps to mitigate hallucinations to some extent but this is already known and established by previous research.
4. The paper is poorly written with grammar mistakes, typos, poor formatting. A figure (Figure 2) is also missing from the paper, in place of that a blank box appears.

**Questions:**

Suggestions:
1. Authors should improve the formatting of the paper. For example, references are not in parenthesis (e.g., line 33, 35, etc.). Similarly, there are several grammatical mistakes in the paper that authors should fix.

---

### Author Response · Authors · 2024-11-27
**Clarifications on Key Innovations and Figure Placement, with Improvements Addressing Reviewer Feedback**

Thanks for taking the valuable time to review the paper.

First, the main innovation of this paper is its focus on K-12 questions. It also differs from CRAG in the following two key aspects:

(1) Dataset focus and question type:
This dataset is designed for K-12 education use cases, and the questions are at an exam-level standard, evaluating how well students have mastered specific knowledge. In contrast, CRAG uses the DBpedia knowledge graph from Wikipedia (Sun et al., 2024; Yang et al., 2024). The questions in CRAG are not suited for exams. For example, a typical CRAG question is: How many Oscar awards did Meryl Streep win?

(2) Conclusion differences:
The conclusions of our study significantly differ from CRAG. Our research demonstrates that a basic RAG solution effectively mitigates the hallucination problem, whereas CRAG shows limited effectiveness of basic RAG solutions. For instance:

- In CRAG, the basic RAG solution using the Llama 3 70B Instruct model only marginally improved the total score from 3.4% to 7.7% in Task 1 (5 web pages per question).
Additionally, CRAG reports that the basic RAG solution using GPT-4 decreased the score from 20.0% to 7.7% in Task 1 (5 web pages per question).
- However, in our dataset, we observe substantial improvements with the Llama 3 8B model, where the RAG solution increased the total score from 40.24% to 48.53%.

Second, we would like to clarify two questions:
- Reviewer uMw2 and Reviewer C9X8 asked about the position of Figure 2. We would like to clarify that Figure 2 (Benchmark Construction Process) is already included in the paper on Page 4 due to LaTeX formatting. The hyperlink for Figure 2 required adjustment in the previous version, which may have caused difficulty in locating it.
- Reviewer C9X8 asked about the difference of Table 1 and Table 2. The two tables are different. Table 2 is computed based on the numbers in Table 1. While Table 1 shows the absolute numbers (correct answers, incorrect answers, missing answers), Table 2 shows the percentages. We also added explanations in the paper to further elaborate on the two tables.

Third, we made the following improvements:

- As requested by Reviewer uMw2, we addressed formatting and grammatical comments, and updated the reference style.

References:
1. Sun, Kai, et al. "Head-to-Tail: How Knowledgeable are Large Language Models (LLMs)? AKA Will LLMs Replace Knowledge Graphs?." Proceedings of the 2024 Conference of the North American Chapter of the Association for Computational Linguistics: Human Language Technologies (Volume 1: Long Papers). 2024.

2. Yang, Xiao, et al. "CRAG--Comprehensive RAG Benchmark." arXiv preprint arXiv:2406.04744 (2024).

---

### Note · Authors · 2024-12-04

**Comment:**

We sincerely appreciate the reviewers' insightful and constructive comments. After careful consideration, we have decided to withdraw our submission from ICLR. Nonetheless, we included a short summary to address the key questions. During the rebuttal period, we provided clarifications on figures and tables and revised the paper, but we did not receive further feedback. We acknowledge that there is still room to fully address the reviewers' valuable input in the future. We believe this work offers meaningful insights into the application of RAG methods in the education domain, and we look forward to refining it further.

**Withdrawal Confirmation:**

I have read and agree with the venue's withdrawal policy on behalf of myself and my co-authors.